# Using statistical modelling and machine learning in detecting bone properties: A systematic review protocol

Osama Abdelhay[1,*], Rand Alshoubaki[1], Sana Murad[1], Omar Abdel-Hafez[1], Qusai Abdelhay[2], Bassem Haddad[3], Tasneem Alhosanie[3], Hala Ajlouni[3], Leanne Ajlouni[3], Tareq Qarain[3], Hamzeh Murad[3], Taghreed Altamimi[4]

1 Department of Data Science and Artificial Intelligence, Princess Sumaya University for Technology, Amman, Jordan, 2 Department of Orthopaedic Surgery, Al-Bashir Hospital, Amman, Jordan, 3 Division of Orthopaedic, Department of Special Surgery, School of Medicine, The University of Jordan, Amman, Jordan, 4 Software Engineering Department, Alfaisal University, Riyadh, Saudi Arabia

☉ These authors contributed equally to the manuscript.
* osamaabdelhay@gmail.com

## Abstract

### Introduction

Osteoporosis, a common condition characterised by decreased bone mass and microarchitectural deterioration, leading to increased fracture risk, is a significant health concern. Traditional diagnostic methods, such as Dual-energy X-ray Absorptiometry (DXA), have limitations in sensitivity and accessibility. However, the emergence of artificial intelligence (AI) and machine learning (ML) has brought promising tools capable of analysing complex medical data to enhance the detection and prediction of osteoporosis-related bone properties. This systematic review protocol outlines the methodology to evaluate the application and effectiveness of AI and ML methods in detecting bone properties and osteoporosis. It underscores their potential to revolutionise healthcare by providing more accurate and accessible osteoporosis detection and prediction tools.

### Methods

This systematic review, which will follow the Preferred Reporting Items for Systematic Reviews and Meta-Analysis Protocols (PRISMA-P) guidelines, will be comprehensive in its approach. A thorough search will be conducted across PubMed, Embase, IEEE Xplore, Scopus, Cochrane Library, and GitHub from their inception to March 2025. Studies involving adults aged 40 years and older that utilise AI/ML techniques to detect or predict bone density or other bone-related properties will be included. Two independent reviewers will perform screening, data extraction, and risk of bias assessments using appropriate tools such as RoB 2, ROBINS-I, QUADAS-2, PROBAST, and NOS. The comprehensive nature of this review ensures that no relevant study is overlooked. Data synthesis will involve narrative synthesis and, if applicable, meta-analysis using Review Manager (RevMan) and R software.

**Data availability statement:** No datasets were generated or analysed during the current study. All relevant data from this study will be made available upon study completion.

**Funding:** The author(s) received no specific funding for this work.

**Competing interests:** The authors have declared that no competing interests exist.

**Abbreviations:** AI, Artificial Intelligence; AUC, Area Under the Curve; BMD, Bone Mineral Density; CT, Computed Tomography; CDSS, Clinical Decision Support Systems; DXA, Dual-energy X-ray Absorptiometry; EHR, Electronic Health Record; EMA, European Medicines Agency; FDA, Food and Drug Administration; FRAX, Fracture Risk Assessment Tool; ML, Machine Learning; MRI, Magnetic Resonance Imaging; NOS, Newcastle-Ottawa Scale; PRISMA-P, Preferred Reporting Items for Systematic Reviews and Meta-Analysis Protocols; PROBAST, Prediction model Risk of Bias Assessment Tool; PROSPERO, International Prospective Register of Systematic Reviews; QUADAS-2, Quality Assessment of Diagnostic Accuracy Studies 2; RCTs, Randomized Controlled Trials; RevMan, Review Manager; RL, Reinforcement Learning; ROBINS-I, Risk of Bias in Non-randomized Studies - of Interventions; SR, Systematic Review.

## Discussion

This systematic review will comprehensively evaluate current AI and ML applications in detecting bone properties and osteoporosis. By identifying and analysing various AI/ML models and comparing them with traditional diagnostic methods, the review aims to highlight the effectiveness and potential of these technologies in clinical practice. The findings are expected to significantly impact healthcare professionals, researchers, and policymakers regarding advancements in AI/ML for bone health assessment and guide future research directions. Understanding the strengths and limitations of existing studies will be crucial in developing standardised protocols and facilitating the integration of AI/ML tools into routine osteoporosis screening and management.

## Systematic review registration

This Systematic Review Protocol was registered in PROSPERO (CRD42024587326).

## Introduction

### Background and rationale

Machine learning (ML) has emerged as a transformative technology in medicine. It allows users to analyse complex datasets and uncover patterns that enhance diagnostic accuracy, treatment planning, and patient outcomes [1]. By leveraging algorithms that learn from data, ML enables healthcare professionals to make more informed decisions, predict disease progression, and personalise therapies [2]. In medical imaging, ML techniques and intense learning models have significantly improved the interpretation of complex images, aiding in the early detection and characterisation of diseases [3].

In bone health, ML has shown considerable promise in analysing imaging data to assess bone properties such as vertebral fracture load [4], Microarchitecture parameters [5], vertebral height [6], and bone mineral density (BMD) [7]. These critical indicators of bone strength are essential for diagnosing conditions like osteoporosis and predicting fracture risk. Traditional methods of evaluating these parameters can be time-consuming and may need to capture subtle changes in bone quality. ML algorithms can process large volumes of imaging data from modalities like X-ray, computed tomography (CT), and magnetic resonance imaging (MRI) to extract detailed information about bone structure and strength [8]. For instance, ML models have been developed to predict osteoporosis by analysing hip radiographs and estimating BMD, providing a non-invasive and efficient method for early diagnosis [9,10]. Additionally, deep learning algorithms have been utilised to detect vertebral fractures and assess bone microarchitecture, enhancing the ability to identify individuals at high risk of fractures and allowing for timely intervention [8].

### Hypothetical example: AI-assisted Hip X-ray analysis

We provide a hypothetical scenario based on the preliminary scan of the available literature.

### Scenario

A 65-year-old postmenopausal woman, Mrs. X, presents for a routine health check. Her Dual-energy X-ray Absorptiometry (DXA) scan results show a borderline-normal T-score (−1.1), which does not officially qualify her as osteoporotic.

**AI/ML workflow.**

1. **Data Input**: A hip X-ray is taken during her check-up. The digital radiograph is uploaded to an AI/ML system trained on thousands of annotated hip X-ray images.

2. **Feature Extraction**: The AI model automatically segments the femoral head, neck, and trochanteric region. It then quantifies subtle trabecular patterns, cortical thickness, and geometric indices—features that correlate with bone strength [11].

3. **Risk Prediction**: Using a machine learning algorithm (e.g., gradient boosting or convolutional neural network), the system predicts her "microarchitectural risk score." Despite the near-normal T-score, the model identifies microstructural deterioration that raises her fracture-risk score.

4. **Clinical Alert**: The system flags Mrs X as high-risk and recommends further imaging or early intervention, such as nutritional counselling, targeted exercises, or pharmacologic therapy.

5. **Outcome**: Upon follow-up, clinicians confirmed that Mrs. X had early-stage osteopenia with latent risk factors missed by the DXA alone. The AI-driven alert facilitates a proactive management plan, potentially preventing future osteoporotic fractures.

**Practical implications.**

- **Early Detection**: By analysing subtle changes in trabecular architecture, the AI/ML tool catches risk factors earlier than conventional methods [12].

- **Reduced Subjectivity**: Automated feature extraction can minimise human error and inter-reader variability in interpreting X-ray images.

- **Scalable Screening**: Once validated, such an AI workflow could be integrated into routine radiology settings, expanding access to osteoporosis screening in resource-limited areas.

## Objectives

The domain or condition studied in the systematic review is osteoporosis, explicitly focusing on detecting and predicting bone properties related to osteoporosis using artificial intelligence (AI) and machine learning (ML) methods. The primary objective of this systematic review is to identify and evaluate the artificial intelligence (AI) and machine learning (ML) methods used to detect bone properties and assess their effectiveness.

Specific objectives are:

1. To identify the AI and ML techniques applied to individuals at risk of osteoporosis (e.g., postmenopausal women, older adults) for detecting bone density or other bone-related properties.

2. To categorise the types of AI and ML models (e.g., deep learning, neural networks, decision trees) used for diagnosing or predicting osteoporosis based on bone density measurements or related biomarkers.

3. To compare AI/ML-based methods with traditional diagnostic approaches (e.g., Dual-energy X-ray absorptiometry, DXA) regarding sensitivity, specificity, and overall diagnostic accuracy.

4. To evaluate the reported outcomes (e.g., accuracy, precision, recall) of AI/ML models in detecting osteoporosis or abnormal bone properties and to assess the consistency of these outcomes across different studies.

5. To analyse the standard study designs (e.g., cross-sectional, cohort, case-control) utilised in evaluating AI/ML methods for osteoporosis detection.

## Methods

The systematic review protocol followed the Preferred Reporting Items for Systematic Reviews and Meta-analysis Protocol (PRISMA-P) guidelines (S1 File) [13,14]. It has been registered in PROSPERO (CRD42024587326).

### Eligibility criteria

Inclusion criteria:

1. The population at risk for osteoporosis [15–19]:a.  Adults aged 40 years and older.

   b. Postmenopausal women.

   c. Individuals with a history of fractures or at high risk of fractures.

   d. Individuals with risk factors such as low body mass index, smoking, or chronic conditions associated with bone loss.

2. Clinical or research studies involving osteoporosis:

   a. Studies focusing on detecting or predicting bone health (e.g., bone density, bone strength) use artificial intelligence or machine learning methods.

   b. Participants with diagnosed osteoporosis or osteopenia based on diagnostic measures (e.g., DXA scans, bone turnover markers).

   c. Studies include populations with secondary osteoporosis (due to conditions like corticosteroid use or hyperthyroidism) if AI/ML methods are applied for bone health assessment.

Exclusion criteria:

1. Age: Studies involving participants younger than 40 years.

2. Non-human studies: Studies involving animal models or non-human subjects.

3. Non-relevant conditions: Studies focused on osteoporosis-related bone conditions (e.g., osteogenesis imperfecta and bone cancers).

4. Interventions unrelated to AI/ML: Studies that do not involve artificial intelligence or machine learning in assessing or predicting bone properties or osteoporosis risk.

5. Case reports, editorials, and reviews: Exclude studies that are not original research (e.g., narrative reviews, case reports, or opinion pieces).

### Intervention

The interventions or exposures to be reviewed are artificial intelligence (AI) and machine learning (ML) methods applied to detect, predict, or assess bone properties and osteoporosis.

1. Supervised learning algorithms.

2. Unsupervised learning algorithms.

3. Deep learning techniques: Deep learning is a subfield of machine learning that involves training artificial neural networks with many layers (often called "deep" networks) on large datasets. Through these layers, deep learning models can automatically learn and extract hierarchical features from raw data (e.g., images, text) and achieve high performance in complex tasks like image recognition and natural language processing [20].

4. Natural language processing is used to analyse osteoporosis-related textual clinical records.

5. Reinforcement learning (RL) approaches in diagnostics or treatment recommendations for bone health: RL is a paradigm in which an agent learns optimal behaviour through trial-and-error interactions within an environment. The agent receives rewards or penalties based on actions and adjusts its strategy to maximise long-term cumulative reward [21].

6. Hybrid or ensemble methods combining multiple AI/ML techniques for improved accuracy in bone property assessment:Ensemble methods combine multiple individual models (e.g., decision trees, neural networks) to improve predictive accuracy and robustness compared to a single model. Standard ensemble techniques include bagging (e.g., Random Forest), boosting (e.g., XGBoost), and stacking. By leveraging the diverse strengths of each model, ensemble methods often reduce variance, mitigate bias, and enhance overall performance [22].

   Applications in Bone Health:

1. AI/ML models were developed to predict bone density, bone mineral density (BMD), T-scores, and related measures of bone health.

2. Models designed for detecting early signs of osteoporosis or osteopenia.

3. AI/ML systems are used to predict the risk of fractures based on bone properties.

4. Applications of AI/ML in evaluating the effectiveness of treatments or interventions for osteoporosis based on bone health metrics.

   Data Types Used for AI/ML Analysis:

1. Imaging data used to train AI models for detecting bone abnormalities.

2. Clinical data.

3. Genomic data relevant to bone properties and osteoporosis risk.

4. Wearable sensor data or data from biomechanical studies that inform AI models on bone strength and fracture risk.

   AI/ML Models for Predictive or Diagnostic Purposes:

1. AI/ML methods developed for early diagnosis of osteoporosis or related bone conditions.

2. Predictive models estimate the likelihood of developing osteoporosis or sustaining fractures based on clinical and genetic factors.

## Types of studies to be included

The studies included should primarily be conducted in clinical, healthcare, research, or community health settings where AI/ML methods are applied to diagnose or predict osteoporosis and bone-related conditions.

Studies must involve real-world data and settings relevant to osteoporosis diagnostics, focusing on geographic and demographic diversity to ensure broad applicability. Studies conducted outside clinical or real-world settings or in fields unrelated to bone health will be excluded.

This clear context definition will ensure that the review focuses on relevant studies that apply AI/ML techniques in real-world healthcare environments or research settings related to osteoporosis detection and bone health.

Inclusion Criteria for Study Designs:

1. Randomised Controlled Trials (RCTs): Studies that randomly assign participants to receive either AI/ML-based diagnostic tools or traditional diagnostic methods to assess their effectiveness in predicting or detecting osteoporosis or bone properties.

2. Cohort Studies: Prospective cohort studies that follow a group of individuals over time to assess the predictive accuracy or diagnostic effectiveness of AI/ML methods in detecting osteoporosis or predicting bone fractures.

3. Retrospective cohort studies analysing existing data to compare AI/ML models with traditional diagnostic techniques.

4. Cross-sectional Studies

5. Diagnostic Accuracy Studies

6. Studies focused on evaluating the sensitivity, specificity, positive predictive value, and negative predictive value of AI/ML models in diagnosing osteoporosis or predicting fracture risk.

7. Comparisons between AI/ML diagnostic tools and established gold-standard methods like DXA or FRAX.

8. Real-world Evidence Studies: observational studies that assess the application of AI/ML techniques in clinical practice, focusing on their utility and effectiveness in real-world settings.

## Comparator

The AI/ML interventions will be compared to traditional diagnostic methods or control groups. Alternatives could include conventional diagnostic techniques, predictive models, or standard clinical practices for detecting and predicting osteoporosis and bone properties.

## Search strategy

Databases to be Searched.

- PubMed/MEDLINE

- Embase

- IEEE Xplore

- Scopus

- Cochrane Library

- GitHub

Search Terms and Strategy.

The search strategy will combine Medical Subject Headings (MeSH) and free-text terms related to artificial intelligence, machine learning, and bone health. Boolean operators (AND, OR, NOT) and truncation will refine the search. An example of a search strategy for PubMed can be found in the S2 File.

## Search limits

- **Language**: Literature will only be restricted to English-language publications, primarily due to resource constraints (e.g., limited access to translation services and reviewers' expertise in other languages).

- **Publication Date**: All studies published up to December 2024.

- **Study Types**: Filters will be applied to include only original research articles (e.g., clinical trials, cohort studies, diagnostic accuracy studies) and exclude reviews, editorials, and case reports.

## Additional search methods

- **Reference Lists**: Reference lists of included studies and relevant reviews will be manually searched to identify additional studies.

- **Grey Literature**: Conference proceedings, dissertations, and other grey literature sources will be considered to minimise publication bias.

## Data management

- All search results will be imported into reference management software (e.g., EndNote [23] or Zotero [24]) for record-keeping and duplicate removal.

- Two independent reviewers will screen, with a third reviewer resolving discrepancies.

### Study selection

a. Identification of Studies: A comprehensive search will be conducted across multiple databases (e.g., PubMed, Embase, IEEE Xplore, Scopus) using a predefined search strategy based on the keywords artificial intelligence, machine learning, osteoporosis, bone, prediction, detection, and bone properties.

b. Screening of Studies: Title and abstract screening: Two independent reviewers will screen the titles and abstracts of all identified studies to assess relevance based on the inclusion and exclusion criteria. Studies unrelated to AI/ML applications for osteoporosis or bone health will be excluded.

c. Full-text review: Both reviewers will retrieve and review full-text articles of potentially eligible studies. Disagreements will be resolved through discussion or, if necessary, by a third reviewer.

d. Selection Documentation: All decisions made during the study selection process (e.g., reasons for exclusion) will be documented in a PRISMA flow diagram to provide transparency regarding the number of studies included and excluded at each stage.

### Data to be Extracted:

a. General Study Information:

- Study title.

- Authors.

- Year of publication.

- Journal or source of publication.

- Study setting.

b. Study Design and Population:

- Study design.

- Sample size and characteristics.

- Inclusion and exclusion criteria of the study population.

- Geographic location and ethnic diversity of the study population.

c. Intervention/Exposure:

- Type of AI/ML model used.

- Specific algorithms applied.

- Data types used.

- Outcome variables the AI/ML models aimed to predict or diagnose.

d. Comparator:

- Details of the comparison group or alternative method.

- Control group characteristics (if applicable).

e. Outcomes:

- Diagnostic accuracy metrics.

- Predictive accuracy metrics.

- BMD estimation accuracy.

- Fracture risk prediction outcomes.

- Clinical decision-making outcomes.

f. Study Quality and Risk of Bias:

- Risk of bias assessment using appropriate tools.

- Study funding sources and potential conflicts of interest.

**Risk of bias assessment.** This systematic review will assess the risk of bias using the following tools:

1. Cochrane Risk of Bias Tool (RoB 2) for randomised controlled trials (RCTs), assessing bias in randomisation, blinding, outcome data, and reporting [25].

2. ROBINS-I for non-randomized studies, covering biases in confounding, participant selection, and outcome measurement [26].

3. QUADAS-2 for diagnostic accuracy studies, focusing on patient selection, the index test (AI/ML), and reference standards [27].

4. PROBAST for AI/ML model development studies, assessing bias in participants, predictors, outcomes, and analysis [28].

5. Newcastle-Ottawa Scale (NOS) for cohort and case-control studies, assessing participant selection, comparability, and outcome [29].

Two independent reviewers will perform the assessments, with a third reviewer resolving discrepancies. Each study will be rated as low, moderate, or high risk of bias. The results will be documented and used in data synthesis, with sensitivity analyses to explore the effect of excluding studies with a high risk of bias.

## Data synthesis

For this systematic review, we will use narrative synthesis and meta-analysis (if applicable), depending on the nature and availability of data from the included studies.

1. Narrative Synthesis [30]

- Qualitative data (e.g., study characteristics, AI/ML methods, data sources, and outcomes) will be summarised to identify trends in using AI/ML for osteoporosis detection and prediction.

- Study outcomes (e.g., diagnostic accuracy, predictive performance) will be grouped based on the type of AI/ML model, data type (imaging, clinical data), and outcome measures (BMD, fracture risk).

- We will compare the AI/ML models and traditional methods, highlighting differences in performance, applicability, and context (clinical or research settings).

2. Meta-analysis (if applicable)

- We will perform a meta-analysis of studies that report comparable outcome metrics (e.g., sensitivity, specificity, AUC for diagnostic accuracy) [31].

- Given the likelihood of differences in AI/ML models, populations, and settings, a random-
effects model will account for potential variability between studies (heterogeneity) [32].

- Heterogeneity will be assessed using the $I^2$ statistic and Cochran's Q test. Thresholds of $I^2$ will guide interpretation: 25% (low), 50% (moderate), and 75% (high heterogeneity). Meta-regression will be considered if substantial heterogeneity exists to explore factors such as AI/ML model type, data used, or study design [32].

3. Subgroup and Sensitivity Analyses

Subgroup analyses will explore potential effect modifiers such as:
- Type of AI/ML model (e.g., deep learning vs. traditional ML).

- Data source (imaging vs. clinical data).

- Study population (general population vs. high-risk groups).

- Sensitivity analyses will exclude studies with a high risk of bias or those using different definitions of outcomes (e.g., diagnostic thresholds).

4. Software

Review Manager (RevMan) will be used for data management and narrative synthesis [33].
R will be used for the meta-analysis, particularly for generating pooled estimates and forest plots [32]. This approach allows for a comprehensive synthesis of both qualitative and quantitative data, ensuring that AI/ML models' effectiveness is evaluated in the context of osteoporosis and bone health assessment.

### Plan for ongoing updates ("Living systematic review")

We intend to update this systematic review regularly to address the rapid evolution of AI/ML technologies. Specifically, we will employ a "living systematic review" model [34] wherein the search strategy is periodically re-run—at least every 12 months—to identify newly published studies relevant to AI/ML in osteoporosis detection and prediction. To facilitate this, we will:

1. **Set Up Automated Alerts**: We will use alerts from databases such as PubMed's My NCBI, Scopus, and Embase to receive notifications of new publications that match our search criteria.

2. **Screen and Incorporate New Evidence**: Any studies meeting the inclusion criteria will be screened, assessed for risk of bias using the same methodology, and incorporated into updated analyses.

3. **Version Tracking**: We will maintain a transparent version history, clearly stating any changes in eligibility criteria, search terms, or analysis methods.

4. **Periodic Summaries**: Findings from updated literature searches will be summarised and made publicly available, ensuring clinicians, researchers, and policymakers have continuous access to the most current evidence.

By integrating these measures, we aim to keep our review relevant and reflective of the latest innovations in AI/ML for bone health assessment. This strategy will help mitigate the limitations of rapidly evolving methods and ensure our conclusions remain as up-to-date as possible.

## Discussion

Osteoporosis is a significant public health concern characterised by reduced bone mass and deterioration of bone microarchitecture, leading to an increased risk of fractures [35]. Traditional diagnostic methods, such as Dual-energy X-ray Absorption Spectroscopy (DXA), are considered the gold standard but have limitations, including accessibility, cost, and the inability to capture subtle changes in bone quality [36]. Advances in artificial intelligence (AI) and machine learning (ML) offer promising alternatives that could enhance early osteoporosis detection and risk assessment.

This systematic review protocol outlines a comprehensive plan to evaluate the application and effectiveness of AI and ML methods in detecting bone properties and osteoporosis. By systematically identifying and synthesising current research, the review aims to:

- **Bridge Knowledge Gaps**: Provide a detailed overview of existing AI/ML techniques used in bone health assessment, highlighting their capabilities and limitations.

- **Inform Clinical Practice**: Compare AI/ML-based methods with traditional diagnostic approaches to assess their potential integration into clinical workflows.

- **Guide Future Research**: Identify areas where further investigation is needed, such as standardisation of AI/ML models, validation in diverse populations, and real-world applicability.

### Anticipated challenges

- **Heterogeneity of Studies**: Variability in study designs, AI/ML models, and outcome measures may pose challenges in data synthesis and comparison.

- **Risk of Bias**: The included studies' differing quality and potential biases necessitate careful assessment to ensure reliable conclusions.

- **Rapid Technological Advancements**: The fast-paced evolution of AI/ML technologies may result in the emergence of new studies during the review process, requiring updates to the search strategy.

## Strengths and limitations

### Strengths

- **Comprehensive Approach**: Including multiple databases and grey literature aims to capture a wide range of studies and minimise publication bias.

- **Robust Methodology**: Adherence to PRISMA-P guidelines and the use of established risk of bias tools will enhance the reliability and transparency of the review.

- **Interdisciplinary Team**: Collaboration among experts in data science, artificial intelligence, orthopaedics, and software engineering will enrich the analysis and interpretation of findings.

### Limitations

- **Potential Heterogeneity:** Variations in AI/ML models, study designs, and outcome measures may limit the ability to perform a meta-analysis and generalise findings.

- **Data Availability:** Limited reporting or lack of access to complete datasets in some studies may affect the depth of the analysis.

- **Language Constraints:** Restricting the search process to the English language may introduce language bias and limit the generalizability of this review. As language bias can exclude potentially significant findings published in other languages, we aim to incorporate studies—particularly those in Chinese and Spanish—in future updates where possible.

## Potential impact

The findings from this systematic review are expected to contribute valuable insights into the role of AI and ML in osteoporosis detection and management. By identifying effective AI/ML models and highlighting their comparative advantages, the review could support healthcare professionals in adopting innovative technologies that improve patient outcomes. Additionally, the review may stimulate further research and development in this field, ultimately advancing bone health assessment and reducing the burden of osteoporosis-related fractures.

### Influence on clinical workflows

Integrating AI/ML tools into current osteoporosis management pathways offers significant potential to optimise clinical workflows. For instance, automated algorithms embedded in radiology software could swiftly analyse bone density metrics from routinely acquired imaging (e.g., X-rays or CT scans), flagging high-risk patients earlier than standard protocols. Additionally, AI-driven clinical decision support systems (CDSS) can be embedded within electronic health records (EHRs) to synthesise patient-level risk factors—such as age, comorbidities, and lifestyle behaviours—and generate personalised alerts or treatment suggestions. These strategies could reduce clinician workload, improve diagnostic consistency, and streamline patient triage. However, successful implementation will require robust validation studies, user-friendly interfaces, and interprofessional collaboration to ensure workflow integration does not overburden providers or compromise patient safety [37].

### Public health strategies

On a population level, AI/ML-based screening programs can reach a broader segment of at-risk individuals—particularly in resource-limited settings where access to DXA is often constrained. By leveraging remote imaging modalities, mobile health applications, or wearable sensor data, public health agencies could deploy AI-enhanced screening initiatives prioritising high-risk demographics (e.g., older adults, postmenopausal women, and individuals with comorbidities). Such programs may enable earlier intervention, reduce fracture rates, and ultimately decrease the healthcare costs associated with osteoporotic complications. However, ethical considerations related to data privacy and equitable access must be carefully addressed before scaling these technologies nationally or internationally [37,38].

### Guideline development and regulatory considerations

As AI/ML methodologies evolve, clinical guidelines for osteoporosis care must incorporate evidence-based recommendations on algorithm selection, model interpretability, and performance metrics (e.g., sensitivity, specificity, area under the curve). Regulatory bodies and professional organisations—such as the Food and Drug Administration (FDA) in the United States or the European Medicines Agency (EMA) in Europe—may also establish standards for validating and approving AI tools used in bone health assessment. Systematic reviews like this one can provide an evidence base to inform these evolving standards, highlighting models that demonstrate robust predictive power and clinical benefit. Encouraging open science practices (e.g., publicly available model code and validation datasets) can further support guideline development by facilitating transparent peer review and reproducibility of AI/ML approaches [39].

### Future directions

Beyond outlining best practices for initial uptake, future research should explore cost-effectiveness analyses and longitudinal real-world evidence that track patient outcomes post-implementation. Such data could guide ongoing policy refinements, especially as new AI/ML models emerge or existing ones are retrained with larger, more diverse datasets. In this way, the knowledge generated from our systematic review will inform current clinical practice and continually shape the evolution of osteoporosis care in the face of rapidly advancing AI/ML technologies.

### Supporting information

**S1 File. PRISMA-P.**
(DOCX)

**S2 File. Examples of search queries used on different databases.**
(DOCX)

### Author contributions

**Conceptualization:** Osama Abdelhay, Sana Murad.

**Investigation:** Osama Abdelhay, Rand Alshoubaki, Tareq Qarain.

**Methodology:** Osama Abdelhay, Rand Alshoubaki, Sana Murad, Omar Abdel-Hafez, Qusai Abdelhay.

**Project administration:** Sana Murad, Tasneem Alhosanie.

**Resources:** Rand Alshoubaki, Sana Murad, Omar Abdel-Hafez, Bassem Haddad, Tasneem Alhosanie, Hala Ajlouni, Leanne Ajlouni, Tareq Qarain, Hamzeh Murad.

**Software:** Taghreed Altamimi.

**Supervision:** Osama Abdelhay, Qusai Abdelhay.

**Validation:** Leanne Ajlouni, Hamzeh Murad.

**Writing – original draft:** Osama Abdelhay, Rand Alshoubaki, Sana Murad, Omar Abdel-Hafez.

**Writing – review & editing:** Bassem Haddad, Taghreed Altamimi.

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
