## [Decision Letter · Decision Letter 0]

18 Nov 2024

PONE-D-24-42415

Using Statistical Modelling and Machine Learning in Detecting Bone Properties: A Systematic Review Protocol

PLOS ONE

Dear Dr. Abdelhay,

Thank you for submitting your manuscript to PLOS ONE. After careful consideration, we have decided that your manuscript does not meet our criteria for publication and must therefore be rejected.

**Specifically:**

**the work presents a plan on how activities will be carried out to produce a review paper. Unfortunately, I do not believe this kind of study falls within the "Systematic reviews and meta-analyses" type of submission guidelines. **

**In addition, the presented protocol turn out to have already been published on PROSPERO.**

I am sorry that we cannot be more positive on this occasion, but hope that you appreciate the reasons for this decision.

Kind regards,

Alessandra Aldieri

Academic Editor

PLOS ONE

Reviewers' comments:

Reviewer's Responses to Questions

**Comments to the Author**

1. Does the manuscript provide a valid rationale for the proposed study, with clearly identified and justified research questions?

Reviewer #1: Yes

2. Is the protocol technically sound and planned in a manner that will lead to a meaningful outcome and allow testing the stated hypotheses?

Reviewer #1: Yes

3. Is the methodology feasible and described in sufficient detail to allow the work to be replicable?

Reviewer #1: Yes

4. Have the authors described where all data underlying the findings will be made available when the study is complete?

Reviewer #1: No

5. Is the manuscript presented in an intelligible fashion and written in standard English?

Reviewer #1: Yes

6. Review Comments to the Author

You may also provide optional suggestions and comments to authors that they might find helpful in planning their study.

**Reviewer #1: ** The subject of the proposed systematic review is the use of machine learning and artificial intelligence in general for the prediction of bone properties and subsequently fracture risk. It is indeed a very interesting research, but I think the journal should publish the review results, not just the protocol, that is already published on PROSPERO dedicated portal, anyway.

7. PLOS authors have the option to publish the peer review history of their article (what does this mean? ). If published, this will include your full peer review and any attached files.

**Do you want your identity to be public for this peer review?** For information about this choice, including consent withdrawal, please see our Privacy Policy .

Reviewer #1: No

- - - - -

---

## [Decision Letter · Decision Letter 1]

28 Jan 2025

PONE-D-24-42415R1Using Statistical Modelling and Machine Learning in Detecting Bone Properties: A Systematic Review ProtocolPLOS ONE

Dear Dr. Abdelhay,

Thank you for submitting your manuscript to PLOS ONE. After careful consideration, we feel that it has merit but does not fully meet PLOS ONE’s publication criteria as it currently stands. Therefore, we invite you to submit a revised version of the manuscript that addresses the points raised during the review process. Please see the comments of the rewiwers bellow. 

We look forward to receiving your revised manuscript.

Kind regards,

Melissa Orlandin Premaor, M.D., Ph.D

Academic Editor

PLOS ONE

Journal Requirements:

Additional Editor Comments (if provided):

Reviewers' comments:

Reviewer's Responses to Questions

**Comments to the Author**

1. Does the manuscript provide a valid rationale for the proposed study, with clearly identified and justified research questions?

Reviewer #2: Yes

Reviewer #3: Yes

2. Is the protocol technically sound and planned in a manner that will lead to a meaningful outcome and allow testing the stated hypotheses?

Reviewer #2: Yes

Reviewer #3: Partly

3. Is the methodology feasible and described in sufficient detail to allow the work to be replicable?

Reviewer #2: Yes

Reviewer #3: Yes

4. Have the authors described where all data underlying the findings will be made available when the study is complete?

Reviewer #2: Yes

Reviewer #3: Yes

5. Is the manuscript presented in an intelligible fashion and written in standard English?

Reviewer #2: Yes

Reviewer #3: Yes

6. Review Comments to the Author

You may also provide optional suggestions and comments to authors that they might find helpful in planning their study.

Reviewer #2: 1. The manuscript tackles a critical and timely issue by exploring the application of AI and ML in detecting and predicting osteoporosis-related bone properties. This area of research is highly relevant, as traditional diagnostic methods like DXA face limitations in sensitivity and accessibility, leaving room for innovative technologies to make an impact.

2. The authors have presented a strong and methodologically rigorous protocol that adheres to PRISMA-P guidelines. Their use of established risk-of-bias tools, such as RoB 2 and QUADAS-2, ensures the reliability and transparency of the review. This approach enhances the credibility of the proposed study.

3. A notable strength is the interdisciplinary nature of the protocol, which incorporates a variety of AI/ML methods and data types, including imaging, genomic data, and wearable sensor data. This comprehensive scope allows for a broad understanding of AI/ML applications in osteoporosis detection and prediction.

4. However, the decision to include only English-language studies introduces potential language bias. Considering the global nature of osteoporosis research, this could exclude significant findings published in other languages. Addressing this limitation or providing a clear rationale for this choice would strengthen the manuscript.

5. While the authors acknowledge the challenge of rapidly evolving AI/ML technologies, the protocol does not outline a concrete strategy for updating the review with newly published studies. Including a plan to address this would ensure the review remains relevant and comprehensive over time.

6. The absence of illustrative examples or pilot data is a missed opportunity to provide readers with practical context. Even a hypothetical case or brief summary of existing AI/ML applications in osteoporosis could help clarify the practical implications of the study.

7. The discussion section could be expanded to better address the real-world impact of the review. For example, how could these findings influence clinical workflows, public health strategies, or the development of guidelines for integrating AI/ML into osteoporosis care?

8. While the manuscript is generally well-written and logically structured, some technical terms (e.g., ensemble methods, reinforcement learning) could be simplified or briefly explained to ensure accessibility for a broader audience, including clinicians and policymakers.

9. The authors’ commitment to making all data publicly available upon study completion aligns well with open science principles and enhances the transparency of the work. This is a commendable practice.

10. Overall, this protocol is well-conceived and has the potential to contribute significantly to the field. By addressing limitations such as language bias, providing practical examples, and elaborating on the broader implications of the findings, the manuscript could be further strengthened and have a more impactful reach.

Reviewer #3: Esteemed authors,

your proposal for SR and MA about this issue is meaningful and necessary; I would recommend, however, to narrow your inclusion criteria for adults over 60 years old; this recommendation, although it may limit the amount of studies available for review, it also may produce more meaningful results and avoid bias from analysing very different populations in the same MA. Another solution for this would be to analyse younger adults (< 60 YO) separately from older adults.

7. PLOS authors have the option to publish the peer review history of their article (what does this mean? ). If published, this will include your full peer review and any attached files.

**Do you want your identity to be public for this peer review?** For information about this choice, including consent withdrawal, please see our Privacy Policy .

Reviewer #2: **Yes: ** Krit Pongpirul

Reviewer #3: No

---

## [Author Response · Author response to Decision Letter 1]

30 Jan 2025

Response letter to the reviewers

Reviewer #2 Comments

Comment 1

“The manuscript tackles a critical and timely issue by exploring the application of AI and ML in detecting and predicting osteoporosis-related bone properties. This area of research is highly relevant, as traditional diagnostic methods like DXA face limitations in sensitivity and accessibility, leaving room for innovative technologies to make an impact.”

Response:

• We thank the reviewer for this comment.

Comment 2

“The authors have presented a strong and methodologically rigorous protocol that adheres to PRISMA-P guidelines. Their use of established risk-of-bias tools, such as RoB 2 and QUADAS-2, ensures the reliability and transparency of the review. This approach enhances the credibility of the proposed study.”

Response:

• We thank the reviewer for this comment

Comment 3

“A notable strength is the interdisciplinary nature of the protocol, which incorporates a variety of AI/ML methods and data types, including imaging, genomic data, and wearable sensor data. This comprehensive scope allows for a broad understanding of AI/ML applications in osteoporosis detection and prediction.”

Proposed Revision/Response:

• We thank the reviewer for this comment

Comment 4

“However, the decision to include only English-language studies introduces potential language bias. Considering the global nature of osteoporosis research, this could exclude significant findings published in other languages. Addressing this limitation or providing a clear rationale for this choice would strengthen the manuscript.”

Proposed Revision/Response:

1. Rationale Explanation: We have added a subsection under “Search Strategy”, explaining that we have limited the search to English-language studies due to resource constraints (e.g., translation resources, reviewer expertise).

2. Acknowledgment of Limitation: In the “Discussion” or “Limitations” section, we have emphasised that future reviews could broaden the language scope.

Comment 5

“While the authors acknowledge the challenge of rapidly evolving AI/ML technologies, the protocol does not outline a concrete strategy for updating the review with newly published studies. Including a plan to address this would ensure the review remains relevant and comprehensive over time.”

Proposed Revision/Response:

• We have added a section under the methodology about performing a “living systematic review” approach or scheduling periodic updates (i.e., every 12 months) to capture relevant new AI/ML methods in osteoporosis research.

• We have also noted that automated alerts from databases (e.g., PubMed’s My NCBI) will be set up to track newly published relevant literature.

Elliott JH, Synnot A, Turner T, et al. “Living systematic review: An emerging opportunity to narrow the evidence–practice gap.” PLoS Medicine, 14(1), e1002280, 2017. https://doi.org/10.1371/journal.pmed.1002280 

Comment 6

“The absence of illustrative examples or pilot data is a missed opportunity to provide readers with practical context. Even a hypothetical case or brief summary of existing AI/ML applications in osteoporosis could help clarify the practical implications of the study.”

Proposed Revision/Response:

• We incorporated a concise subsection briefly describing an example use case (hypothetical based on the literature) demonstrating how AI/ML might identify subtle bone mineral density changes from hip X-rays.

Comment 7

“The discussion section could be expanded to better address the real-world impact of the review. For example, how could these findings influence clinical workflows, public health strategies, or the development of guidelines for integrating AI/ML into osteoporosis care?”

Proposed Revision/Response:

• We expanded the “Discussion” section to include specific points on how clinicians could implement AI/ML tools in routine workflows (e.g., automated BMD measurements in radiology reports and integration with electronic health records).

• We discussed broader public health implications, including how predictive modelling might guide population screening strategies or resource allocation in ageing populations.

• We added a subsection about the future directions and the impact of systematic reviews such as ours on establishing evidence-based practices.

Comment 8

“While the manuscript is generally well-written and logically structured, some technical terms (e.g., ensemble methods, reinforcement learning) could be simplified or briefly explained to ensure accessibility for a broader audience, including clinicians and policymakers.”

Proposed Revision/Response:

• We added short footnotes to explain key AI/ML technical terms (e.g., ensemble methods, reinforcement learning, deep learning) in non-technical language.

Comment 9

“The authors’ commitment to making all data publicly available upon study completion aligns well with open science principles and enhances the transparency of the work. This is a commendable practice.”

Proposed Revision/Response:

• We thank the reviewer for this comment

Comment 10

“Overall, this protocol is well-conceived and has the potential to contribute significantly to the field. By addressing limitations such as language bias, providing practical examples, and elaborating on the broader implications of the findings, the manuscript could be further strengthened and have a more impactful reach.”

Proposed Revision/Response:

• We thank the reviewer for this comment. We believe the comments we received have enhanced our protocol. We hope the modifications the manuscript underwent align with the reviewer's comment.

Reviewer #3 Comments

Comment

“Your proposal for SR and MA about this issue is meaningful and necessary; I would recommend, however, to narrow your inclusion criteria for adults over 60 years old; this recommendation, although it may limit the amount of studies available for review, it also may produce more meaningful results and avoid bias from analysing very different populations in the same MA. Another solution for this would be to analyse younger adults (< 60 YO) separately from older adults.”

Proposed Revision/Response:

• We thank the reviewer for this comment. To reduce the bias concern that the reviewer mentioned, we mentioned in the methodology that sensitivity and stratified analysis would be considered to account for potential sources of bias and heterogeneity. Moreover, given enough power and sample size, i.e., the number of studies and subjects within the studies, we aim to produce a meta-regression to reduce confounding effects.

---

## [Editor Report · Decision Letter 2]

5 Feb 2025

Using Statistical Modelling and Machine Learning in Detecting Bone Properties: A Systematic Review Protocol

PONE-D-24-42415R2

Dear Dr. Abdelhay,

We’re pleased to inform you that your manuscript has been judged scientifically suitable for publication and will be formally accepted for publication once it meets all outstanding technical requirements.

Kind regards,

Melissa Orlandin Premaor, M.D., Ph.D

Academic Editor

PLOS ONE
---

## [Editor Report · Acceptance letter]

PONE-D-24-42415R2

PLOS ONE

Dear Dr. Abdelhay,

I'm pleased to inform you that your manuscript has been deemed suitable for publication in PLOS ONE. Congratulations! Your manuscript is now being handed over to our production team.

Kind regards,

on behalf of

Dr. Melissa Orlandin Premaor

Academic Editor

PLOS ONE